# Mechanical and Biomimetic Characteristics of Bulk-Fill Resin Dental Composites Following Exposure in a Simulated Acidic Oral Environment

**DOI:** 10.3390/biomimetics8010019

**Published:** 2023-01-04

**Authors:** Waheed Murad Dahri, Naresh Kumar, Noorulain Altaf, Waqas Mughal, Muhammad Sohail Zafar

**Affiliations:** 1Medical Research Centre, Liaquat University of Medical and Health Sciences, Jamshoro 76090, Pakistan; 2Department of Science of Dental Materials, Bibi Aseefa Dental College, Shaheed Mohtarama Benazir Bhutto Medical University, Larkana 77150, Pakistan; 3Department of Science of Dental Materials, Dr. Ishrat Ul Ebad Khan Institute of Oral Health Sciences, Dow University of Health Sciences, Karachi 74200, Pakistan; 4Department of Mechanical Engineering, Quaid-e-Awam, University of Engineering, Science and Technology, Nawabshah 67480, Pakistan; 5Department of Restorative Dentistry, College of Dentistry, Taibah University, Al Madina 41311, Saudi Arabia; 6Department of Dental Materials, Islamic International Dental College, Riphah International University, Islamabad 44000, Pakistan

**Keywords:** acidic beverages, biomimetics, bulk-fill, nanohybrid, flexural strength, flexural modulus, resin-based composites, storage conditions, surface hardness

## Abstract

During the last 10 years, various companies have marketed different “bulk-fill” resin dental composites for the restoration of posterior stress-bearing teeth; however, the impact of acidic conditions on these relatively newer materials has not been thoroughly investigated. Therefore, an attempt was made to evaluate the effect of acidic beverages on the mechanical biomimetic characteristics of four bulk-fill and one conventional nanohybrid resin-based dental composites (RBCs). The specimens of each RBC were stored in two acidic beverages namely ‘Orange Juice’ and ‘Coca-Cola’, whereas ‘dry’ and ‘distilled water’ storage of specimens served as controls. After 1 week of storage, flexural and surface hardness properties of specimens were determined using a universal testing machine and Vickers hardness tester, respectively. In general, the ‘Coca-Cola’ beverage caused the greatest degradation of flexural strength, flexural modulus, and surface hardness characteristics in all RBCs in contrast to the ‘dry’, ‘distilled water’ controls and ‘Orange Juice’ storage conditions. However, the overall mechanical biomimetic performance of nanohybrid RBCs was relatively better than all other bulk-fill RBCs and may, therefore, be considered a suitable candidate for the restoration of posterior stress-bearing permanent dentition.

## 1. Introduction

The light-cured resin-based dental composite restorative materials are commonly utilized in deep and large cavities worldwide [1,2]. Multiple incremental layers are needed for deeper and larger cavities due to the limited depth of cure of these materials [3,4]. Moreover, the likelihood of shrinkage stress is also minimized [5]. However, the layering technique and subsequent curing shots for resin-based composites (RBCs) polymerization are time-consuming. Consequently, the demand from clinicians for the provision of RBCs with faster and easier procedures has increased. This demand is likely to be met by the development of RBCs with shorter curing times as well as deeper light penetration. 

During the last 10 years, various companies have marketed different ‘‘bulk-fill’’ RBCs. It is claimed by the manufacturers that this class of material could be cured up to 4 mm; as a result, time could be saved. Moreover, manufacturers have also highlighted that this class of materials exhibits lower polymerization shrinkage compared to flowable and conventional RBC counterparts [6]. Consequently, complications associated with polymerization shrinkages [7] such as secondary caries [8,9], postoperative sensitivity, pulpal irritation [10], or cusp deflections [11,12] are likely to be reduced.

The low shrinkage stress of bulk-fill RBCs has been attributed to the altered filler content or resin matrix, whereas the increased depth of cure for bulk-fill RBCs is probably because of their greater translucency [13]. Bulk-fill RBCs usually possess reduced filler content and relatively larger filler particles [14] to facilitate their deeper curing. Many commercially available bulk-fill RBCs, for instance, SureFil SDR flow, x-tra fil, and SonicFill are composed of filler particles larger than 20 μm [14,15]. As a result, the total resin–filler interface decreases, which in turn reduces the scattering of light and enhances the penetration of blue light. One manufacturer has incorporated a germanium-based initiator in one bulk-fill RBC in addition to camphorquinone (CQ) [16]. It is believed that this new initiator exhibits a greater light penetration compared to CQ owing to its higher absorption in the light spectrum ranging from 400 to 450 nm [16].

The SureFil^®^ SDR™ (Smart Dentin Replacement), the first-introduced bulk-fill material, comprises a polymerization modulator that possesses a high molecular weight and is chemically surrounded by the polymerizable resin backbone of the SDR™ monomer. The modulator is believed to provide optimum flexibility to the SDR™ resin [17]. Researchers have reported significantly lower shrinkage stress of RBCs with SDR™ technology [18] in contrast to the flowable, hybrid, and nanofilled RBCs.

These materials are available in various viscosities and could be used as flowable base materials, which need 2 mm of a conventional hybrid RBC as a superficial increment or as a final restoration that does not require an outer increment [19,20,21].

Under erosive conditions, RBCs may be damaged due to the degradation of monomers. Various factors, such as the composition of the resin matrix, its chemical bond, hydrophilicity, and the pH of erosive beverages may affect the speed of such degradation [22]. Therefore, the long-term performance of RBCs is mainly associated with their resistance to degradation in an acidic oral environment [23]. 

Nowadays, the biomimetic concept is considered highly significant in the development of restorative dental materials. One of the major goals of biomimetics is to introduce newer restorative materials to clinical dentistry, which are capable of mimicking the natural tooth in terms of biomechanics [24]. From the mechanical biomimetic viewpoint, surface hardness [25,26] and elastic modulus [27,28] of restorative materials are extensively investigated to foresee their performance in the real clinical environment.

As far as bulk-fill RBCs are concerned, various investigations have assessed the effect of acidic beverages on the various biomimetic aspects, for instance, surface hardness [29,30,31], roughness [29], color stability [32], and elastic modulus [33] of the bulk-fill RBCs. However, the selection of the bulk-fill RBCs in these studies is limited, and until now, the overall performance of these materials in terms of mechanical biomimetic characteristics such as surface hardness and flexural modulus under simulated similar oral acidic conditions has not yet been investigated. Therefore, this research aimed to evaluate the effect of acidic beverages on the surface hardness, flexural strength, and flexural modulus of four bulk-fill and one nanohybrid RBCs. The null hypothesis established was that the acidic beverages namely ‘Orange Juice’ and ‘Coca-Cola’ would not influence the mechanical performance of bulk-fill RBCs.

## 2. Materials and Methods

A total of five commercially available RBCs were purchased from a local vendor. The details of each RBC are given in Table 1. In addition, two beverages, Coca-Cola (The Coca-Cola Company, Lahore, Pakistan) and Orange Juice (Nestle Pakistan Limited, Lahore, Pakistan) were purchased from a local supermarket in Hyderabad, Pakistan. The pH of Coca-Cola and Orange Juice is 3.54 and 4.95, respectively, as reported previously [34].

### 2.1. Specimen Preparation for the Evaluation of the Three-Point Bending Test

For each RBC, a total of 20 bar-shaped specimens (25 mm × 2 mm × 2 mm) were manufactured (n = 100). For each specimen, the RBC was packed into a stainless steel mold and then each side of the specimen was covered with a piece of 0.1 mm thick acetate sheet to seal the oxygen inhibition layer [35]. Each specimen was polymerized from one side using a curing unit (Elipar LED curing unit, 3M ESPE, Seefeld, Germany) having a 10 mm light guide tip and 1200 mW/cm^2^ irradiance at a temperature of 23 ± 2 °C. Due to the 25 mm length of the bar-shaped specimens, an overlapping polymerizing method was employed, as reported in ISO 4049, 2000 [36]. Firstly, the central part of the specimen was polymerized with the light for 20 s and then two light exposures were made on the specimen at two intersecting positions for 20 s each. Subsequently, the acetate sheet was peeled off and the specimen was immediately detached from the mold. After the removal of the specimen, a sharp blade was used to cut away the flesh and the dimensions of each specimen were measured using a micrometer screw gauge (Moore and Wright, Sheffield, UK). Afterward, specimens of each RBC were immersed in two acidic beverages namely ‘Coca-Cola’ (CC) (n = 5) and ‘Orange Juice’ (OJ) (n = 5). In addition, ten specimens of each RBC were stored dry (DC) (n = 5) and in distilled water (DWC) (n = 5) as controls. For the CC, OJ, and DWC storage conditions, 50 mL of each medium was transferred to a beaker and then the specimen set (n = 5) of each RBC was stored in the corresponding medium. The specimens were placed in such a way that they did not touch each other and so that each specimen received similar exposure to the corresponding medium. To cater to all storage conditions, a temperature of 37 °C was maintained for 1 week. In order to prevent the accumulation of leached ingredients from the RBCs, the storage medium was refreshed every 24 h [37].

### 2.2. Specimen Preparation for the Evaluation of Surface Hardness 

For each RBC, a total of 20 specimens (15 mm diameter and 1 mm thickness) were manufactured for the evaluation of surface hardness (n = 100). For each specimen, the RBC was packed into the stainless steel mold and then each side of the specimen was covered with a piece of 0.1 mm thick acetate sheet to seal the oxygen inhibition layer [35]. Each specimen was polymerized from one side using a curing unit (Elipar LED curing unit, 3M ESPE, Seefeld Germany) having a 10 mm light guide tip and 1200 mW/cm^2^ irradiance at a temperature of 23 ± 2 °C. The specimen was polymerized in an orbital sequence four times for 20 s each in overlapping shots [38]. Subsequently, the acetate sheet was peeled off and the specimen was immediately detached from the mold. After the removal of the specimen, excess material was cut away using a sharp blade. Afterward, specimens of each RBC were immersed in two acidic beverages namely ‘Coca-Cola’ (CC) (n = 5) and ‘Orange Juice’ (OJ) (n = 5). In addition, ten specimens of each RBC were stored dry (DC) (n = 5) and in distilled water (DWC) (n = 5) as controls. For the CC, OJ, and DWC storage conditions, 50 mL of each medium was transferred to a beaker and then the specimen set of each RBC was stored in the corresponding medium. The specimens were placed in such a way that they did not touch each other and so that each specimen received similar exposure to the corresponding medium. For all storage conditions, a temperature of 37 °C was maintained for 1 week. To prevent the accumulation of leached ingredients from the RBCs, the storage medium was refreshed every 24 h [37].

### 2.3. Determination of Flexural Strength and Modulus

After completion of the 1-week storage cycle, the specimens from each corresponding storage condition were removed from the storage medium and tested using a three-Point flexural configuration in a universal testing machine (M500-5CT Testometric, Rochdale, UK). The test was carried out at a cross-head speed of 1 mm/min and the support span length was 20 mm (Figure 1). The maximum load was recorded, and both flexural characteristics were calculated for each specimen using the standard formulas below [39]:

The formula for the calculation of Flexural strength (FS)
σ = 3Fl/2bh^2^
where σ denotes the flexural strength (MPa), F is the maximum load (N) applied to the specimen, l is the space between the supports (mm), and b and h are the width (mm) and height (mm) of the specimen, respectively.

The formula for the calculation of Elastic Modulus (EM)
E = Fl^3^/4bh^3^d
where E is the elastic modulus (GPa), F is the maximum load (N) applied to the specimen, l is the space between the supports (mm), b and h are the width (mm) and height (mm) of the specimens, respectively, and d is the deflection (mm).

### 2.4. Evaluation of Surface Hardness (Vickers Microhardness)

The specimens from each corresponding storage condition were removed from the storage medium and then three indentations were performed for each specimen using a digital Vickers hardness tester (Indentec ZHV, Zwick/Roell Indentec, Brierley Hill, UK). A maximum loading force of 100 g was applied using a diamond indenter for 15 s [34]. After the application of loading force for the specified time, the length of both diagonals for each indentation was selected in the built-in microscope and the Vickers hardness number (VHN) was recorded (Figure 2). 

### 2.5. Statistical Analysis

The data were analyzed by Minitab statistical software (version 19) (Minitab Ltd., Coventry, UK). A one-way analysis of variance (ANOVA) and post hoc Tukey’s test were conducted on the data sets to highlight the differences between means of surface hardness, flexural strength, and flexural modulus following the different storage conditions. Moreover, main effects plots were also generated to obtain further insight into the effect of storage condition and RBC type on the combined surface hardness, flexural strength, and flexural modulus data.

## 3. Results

The flexural strength of each RBC, except for GR, significantly decreased following the immersion in OJ and CC compared to DC and DWC conditions (*p* < 0.05). The GR RBC revealed a stable flexural strength under DC (215.13 MPa), DWC (205.83 Mpa), and OJ (212.42 Mpa) storage conditions; however, a decline in the flexural strength of the GR RBC was observed under the CC (132.50 Mpa) storage condition in contrast to the aforementioned three conditions (*p* < 0.05) (Table 2).

Likewise, the flexural modulus of each RBC, except for the GR RBC, significantly decreased following the immersion in OJ and CC compared to DC and DWC conditions (*p* < 0.05). The flexural modulus of the GR RBC showed insignificant difference under DC (8.50 GPa), DWC (8.30 GPa), and OJ (7.82 GPa) storage conditions (*p* > 0.05); however, a decline in the flexural modulus of the GR RBC was observed under the CC (4.53 GPa) storage condition in contrast to the DC, DWC and OJ storage conditions (*p* < 0.05) (Table 2).

The degradation trends concerning the surface hardness of all RBCs under investigation varied greatly compared to the findings for flexural modulus and flexural strength. Interestingly, the surface hardness of XBF was not affected under different storage conditions (*p* > 0.05). Although surface hardness values of the GR RBC under DC (90.80 VHN) and DWC (85.00 VHN) conditions were greater than all bulk-fill RBCs under similar conditions, the GR RBC revealed a substantial decline in descending order from DC (90.80 VHN) to DWC (85.00 VHN), OJ (79.80 VHN) and CC (72.80 VHN) storage conditions (*p* < 0.05) (Table 2).

The mean and standard deviation values of surface hardness, flexural strength, and flexural modulus data of each RBC following different storage conditions are shown in Table 2.

Following each storage condition, the GR RBC appeared to be significantly stronger since it showed the greatest flexural strength values in contrast to all bulk-fill RBCs (*p* < 0.05) (Table 3). In addition, the flexural modulus of the GR RBC was considerably higher than all bulk-fill RBCs following each storage condition (*p* < 0.05), except for CC, as no statistically significant differences were identified among the flexural moduli of GR (4.53 GPa), XBF (4.64 GPa), and TBF (3.9 GPa) RBCs (*p* > 0.05) (Table 3). The surface hardness of the GR RBC was greatest compared to all of the bulk-fill RBCs following DC and DWC storage conditions (*p* < 0.05); however, no statistically significant variation in surface hardness was identified between the GR and XBF RBCs following OJ and CC storage conditions (*p* > 0.05). The mean surface hardness, flexural strength, and flexural modulus along with standard deviation values of all RBCs following each storage condition are given in Table 3. 

The main effects plots of the flexural strength, flexural modulus, and surface hardness data emphasizing the major effect of storage condition and RBC type are given in Figure 3a–c.

## 4. Discussion

In the current research work, highly relevant mechanical characteristics namely flexural strength, flexural modulus, and surface hardness of four bulk-fill RBCs were assessed following their exposure to an acidic environment. This attempt was made to predict the clinical performance of these materials under similar situations. Overall, the degradation of all RBCs including the GR nanohybrid RBC in terms of flexural strength, flexural modulus, and surface hardness was recorded following their immersion in acidic beverages (Table 2 and Table 3, and Figure 3a–c); hence the null hypothesis was rejected. The decline in the aforementioned properties may be accredited to the softening or degradation of the polymer matrix, loss of inorganic filler particles and the breakdown of the resin–filler interface [40,41,42] The findings of our work regarding the deterioration of mechanical properties of bulk-fill RBCs agree with previous investigations to a great extent [31,33]. In a study by Colombo et al. [31], the microhardness of XBF RBC was evaluated following a one-week immersion in CC acidic drink. The authors observed a significant decline in the microhardness of the CC-exposed specimens compared to the DWC group. Borges et al. [33] evaluated the Vickers hardness, diametral tensile strength, and elastic modulus of XBF RBC specimens after 30 days of immersion in CC beverage and observed a substantial reduction in each property in contrast to the control group. In a recent study, Degirmenci et al. [43] investigated the elastic modulus, microhardness, and flexural strength parameters of the Estelle bulk-fill flow RBC (Tokuyama Dental Corporation, Tokyo, Japan) after 1 day, 7 days, 30 days, and 365 days storage in OJ and CC beverages. According to their findings, all parameters significantly declined after each storage cycle and, in particular, the flexural strength of the material did not meet the ISO 4049 standard following short and long-term immersion cycles. 

Under each storage condition, all bulk-fill RBCs exhibit consistently lower mechanical properties than the GR nanohybrid RBCs (Table 3, Figure 3a–c), hence their application in high-load-bearing occlusal tooth cavities may be questioned. In a previous study, Leprince et al. [13] also identified the higher elastic modulus and surface hardness of the GR nanohybrid RBC compared to the bulk-fill counterparts following 24 h of dry storage conditions. Likewise, Vidhawan et al. [44] have reported lower bi-axial flexural strength of FBF, TBF, and XBL RBCs compared to a conventional Filtek Z250 microhybrid RBC (3M ESPE, St. Paul, MN, USA). Moreover, based on previous flexural strength, elastic modulus, and surface hardness literature regarding RBCs [45,46,47], the GR RBC has been considered the best among various commercial RBCs mainly due to its higher filler quantity. Among all the RBCs evaluated in the current research work, the GR nanohybrid RBC contains the highest filler volume percentage, hence its superior properties may be anticipated. 

Among all bulk-fill RBCs investigated in the current research study, a wide variability can be witnessed for all their mechanical properties, which may also be linked to the distinct filler content. For instance, the XBF RBC exhibited relatively greater values of flexural modulus and surface hardness, and these findings may be correlated with their filler amount since they comprised 70.1 vol% of fillers in contrast to 42.5 vol%, 61 vol%, and 66 vol% for FBF, TBF, and QBF, respectively. Leprince et al. [13] also observed similar trends in the bulk-fill RBCs and confirmed the same after finding a good linear correlation between mechanical characteristics and filler mass percentage (R > 0.8). 

In a previous study [48], the investigators evaluated the influence of CC, OJ, and Red Bull beverages on the surface hardness of five microhybrid and three nanohybrid RBCs. The findings of their study highlighted a significant decline in the surface hardness of all RBCs following the immersion in the abovementioned beverages compared to the distilled water control group. These findings are in agreement with our study since a similar trend was also observed in the nanohybrid RBC following the one-week storage in CC and OJ beverages.

The effect of duration and mode of photo-polymerization on the mechanical characteristics of RBCs is well-evident [49]. However, in our study, the duration and mode of photo-polymerization were consistent for each test; hence other researchers must consider such variables when comparing their results with the findings of this study. 

Although variability in filler content is considered a key factor for the inconsistent mechanical characteristics of bulk-fill RBCs, in addition, the role of the resin matrix, photoinitiator chemistry, and filler particle size cannot be overlooked. In some bulk-fill RBCs, the resin matrix has been altered to lessen the polymerization shrinkage [13], but such alteration may not necessarily favor the mechanical behavior of the material under an acidic environment. Such resins are probably more prone to softening and, hence, may change the failure mechanisms of RBCs. Nevertheless, the presence of relatively larger filler particles in the bulk-fill RBCs facilitates the deeper penetration of curing light [14,15], but their mechanical properties may not be essentially comparable with the conventional microhybrid and nanofilled RBCs. Furthermore, the attempt to incorporate different photoinitiators in the bulk-fill RBC is also well-evident [16] and such attempts are also likely to cause a difference in the mechanical performance of bulk-fill RBCs. 

The FBF RBC revealed significantly greater flexural strength values compared to the QBF RBC following each storage condition, and it was not expected since the former comprises a relatively lower filler content than the latter. The possible explanation for this finding could be the plastic deformation and viscoelastic behavior of the FBF RBC due to the presence of a significantly greater amount of resin matrix. It is believed that such material behavior is likely to slow down crack propagation, which may consequently enhance flexural strength [50]. 

Most manufacturers do not disclose the exact composition of the RBCs and as a result, it is hard to find out the exact constituent responsible for the variability in the properties of the RBCs. It is proposed that more studies should be carried out on the experimental bulk-fill RBCs with specific formulations to better elucidate their mechanical performance more logically. Nonetheless, the findings of the current study are very significant and highlight the good standing of the GR nanohybrid RBC. Keeping the overall mechanical performance in view, the bulk-fill RBCs may not be considered suitable candidates for large load-bearing occlusal cavities. 

The strength of the current study includes the evaluation of four commonly available bulk-fill RBCs following acidic storage conditions, an expansion on previous studies that have included only a limited number of similar RBC types. Moreover, our study considered three well-established and recommended mechanical properties so as to present comprehensive results regarding the bulk-fill RBCs, whereas most of the previous relevant studies have evaluated either one or two mechanical properties following acidic storage conditions. The current research work evaluated the impact of acidic beverages on the mechanical performance of RBCs and did not consider the buffering capacity of saliva; hence this may be considered a limitation of the study. Moreover, RBC specimens were stored in the beverages for 1 week before testing. Consequently, further long-term research studies are warranted regarding the influence of saliva on the acidic effect of beverages. 

## 5. Conclusions

It can be concluded from the above discussion and analysis that the ‘Coca-Cola’ beverage caused a greater degradation in flexural strength, flexural modulus, and surface hardness of all RBCs in contrast to the ‘dry control’, ‘distilled water control’, and ‘Orange Juice’ storage conditions. The performance of the GR nanofilled RBC was substantially better than all bulk-fill RBCs. Therefore, it may be considered a suitable candidate for the restoration of posterior stress-bearing permanent teeth.

## Figures and Tables

**Figure 1 biomimetics-08-00019-f001:**
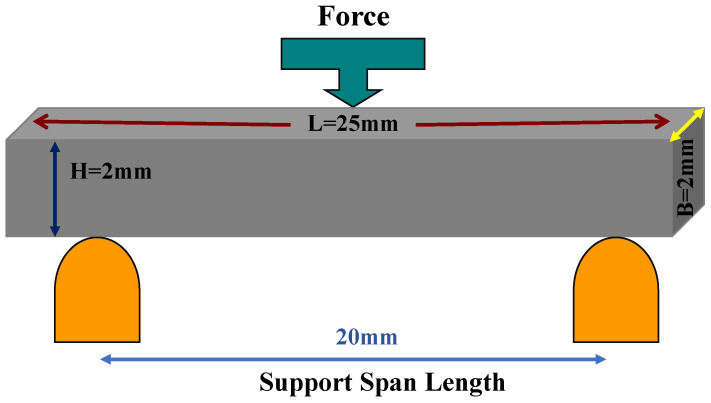
Schematic representation of the three-point flexural test.

**Figure 2 biomimetics-08-00019-f002:**
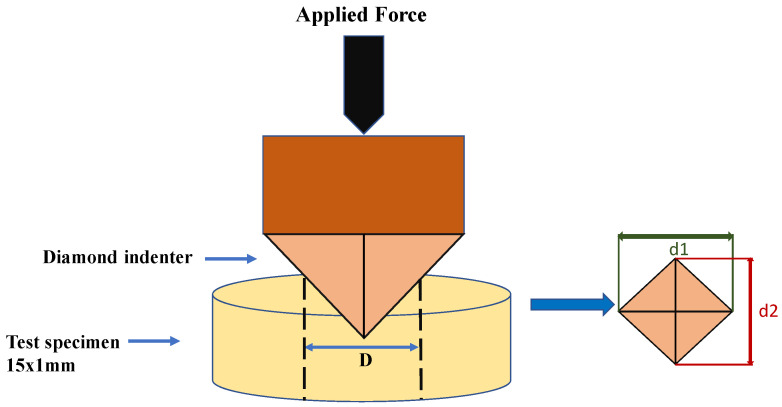
Schematic representation of surface hardness test.

**Figure 3 biomimetics-08-00019-f003:**
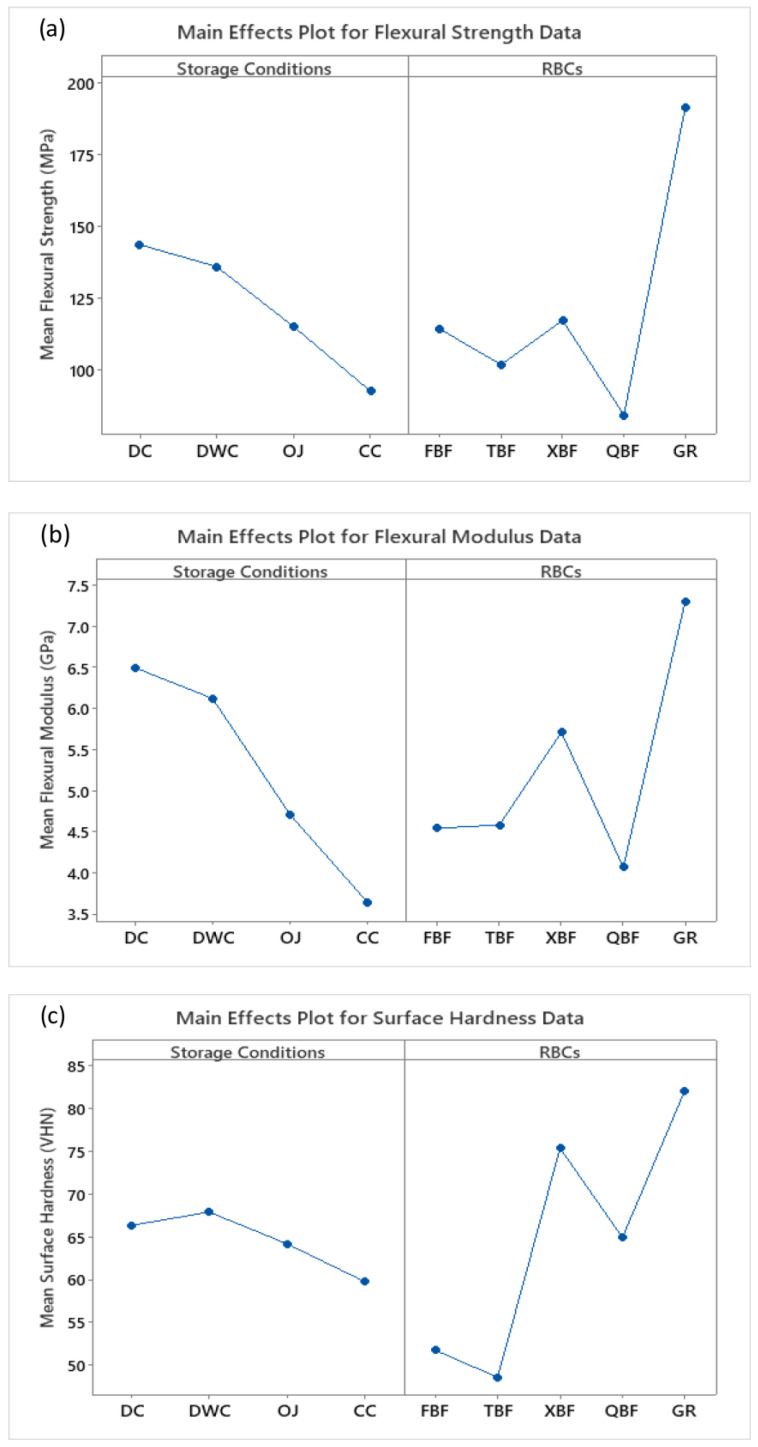
The main effect plots of (**a**) flexural strength, (**b**) flexural modulus, and (**c**) surface hardness data emphasize the major effect of storage condition and RBC type. The specimens stored in the ‘CC’ beverage exhibited the lowest surface hardness, flexural strength, and flexural modulus values compared to other storage conditions. The GR RBC reveals the highest surface hardness, flexural strength, and flexural modulus values in contrast to all bulk-fill RBCs.

**Table 1 biomimetics-08-00019-t001:** The composition of resin-based composites evaluated in the current study.

Resin-Based Composite	Type	Manufacturer	Filler	Filler Weight%; Volume%	Resin Matrix
Filtek Bulk-fill (FBF)	Flowable	3M ESPE, St Paul, MN, USA	Zirconia/silica and ytterbium Trifluoride	64.5; 42.5	Bis-GMA, UDMA, Bis-EMA, and Procrylat resins
Tetric Evoceram Bulk-fill (TBF)	Paste	Vivadent, Schaan, Liechtenstein	Ba-Al-Si-glass, prepolymer filler (monomer, glass filler, and ytterbium fluoride) Spherical mixed oxide	79–81; (including 17% prepolymers); 60–61	Bis-GMA and UDMA
X-tra fil (XBF)	Paste	VOCO (Cuxhaven, Germany)	N/P	86; 70.1	Bis-GMA, UDMA, and TEGDMA
QuiXfil (QBF)	Paste	Dentsply Caulk, Germany	Silinated strontium, aluminum sodium, fluoride phosphate, and silicate glass	86; 66	UDMA, TEGDMA, di- and trimethacrylate resins, and carboxylic acid-modified dimethacrylate resin
Grandio (GR)	Nanohybrid	VOCO (Cuxhaven, Germany)	Al-Si Glass and SiO_2_	87; 71	Bis-GMA, UDMA, and TEGDMA

Abbreviations: Bis-GMA, bisphenol A diglycidyl ether dimethacrylate; UDMA, urethane dimethacrylate; Bis-EMA, bisphenol A polyethylene glycol diether dimethacrylate; TEGDMA, triethylene glycol dimethacrylate; and N/P, information not provided by the manufacturer.

**Table 2 biomimetics-08-00019-t002:** The flexural strength, flexural modulus, and surface hardness of each RBC under different storage conditions.

Materials	Storage Condition	Flexural Strength (MPa) Mean (SD)	Flexural Modulus (GPa) Mean (SD)	Surface Hardness (VHN) Mean (SD)
Filtek Bulk-Fill (FBF)	Dry Control	132.72 (9.28) ^A^	6.20 (0.88) ^A^	50.00 (3.67) ^B^
	Distilled Water Control	126.11 (8.91) ^A^	5.73 (0.38) ^A^	62.00 (6.16) ^A^
	Orange Juice	103.40 (8.36) ^B^	3.54 (0.47) ^B^	47.60 (2.07) ^B^
	Coca-Cola	94.32 (8.25) ^B^	2.68 (0.45) ^B^	47.20 (3.70) ^B^
Tetric EvoCeram Bulk-Fill (TBF)	Dry Control	118.63 (7.76) ^A^	5.44 (0.95) ^A^	43.80 (2.77) ^B^
	Distilled Water Control	109.99 (8.80) ^A^	4.96 (0.53) ^AB^	48.40 (2.88) ^AB^
	Orange Juice	89.14 (7.92) ^B^	3.97 (0.41) ^B^	52.60 (4.16) ^AB^
	Coca-Cola	89.19 (8.93) ^B^	3.95 (0.42) ^B^	49.40 (2.41) ^A^
X-tra fil (XBF)	Dry Control	136.10 (12.46) ^A^	6.86 (0.36) ^A^	77.40 (4.72) ^A^
	Distilled Water Control	128.77 (6.86) ^A^	6.32 (0.76) ^A^	75.80 (4.49) ^A^
	Orange Juice	108.67 (8.42) ^B^	5.01 (0.52) ^B^	75.20 (3.49) ^A^
	Coca-Cola	95.20 (6.67) ^B^	4.64 (0.53) ^B^	73.00 (2.00) ^A^
Quix Fill (QBF)	Dry Control	115.44 (9.58) ^A^	5.48 (0.59) ^A^	69.60 (2.88) ^A^
	Distilled Water Control	109.00 (11.81) ^A^	5.27 (0.70) ^A^	68.20 (2.59^) A^
	Orange Juice	61.22 (8.93) ^B^	3.16 (0.58) ^B^	65.60 (1.14) ^A^
	Coca-Cola	50.73 (4.94) ^B^	2.33(0.34) ^B^	56.20 (3.03) ^B^
Grandio (GR) (Control)	Dry Control	215.13 (8.13) ^A^	8.50 (0.44) ^A^	90.80 (2.16) ^A^
	Distilled Water Control	205.83 (10.05) ^A^	8.33 (0.58) ^A^	85.00 (4.42) ^B^
	Orange Juice	212.42 (7.67) ^A^	7.82 (0.59) ^A^	79.80 (1.64) ^C^
	Coca-Cola	132.50 (7.37) ^B^	4.53 (0.48) ^B^	72.80 (1.92) ^D^

Superscripts with dissimilar letters within the columns for each separate RBC type show statistically significant differences (*p* < 0.05).

**Table 3 biomimetics-08-00019-t003:** The flexural strength, flexural modulus, and surface hardness of all RBCs under each storage condition.

Storage Condition	Materials	Flexural Strength (MPa) Mean (SD)	Flexural Modulus (GPa) Mean (SD)	Surface Hardness (VHN) Mean (SD)
Dry Control	Filtek Bulk-Fill (FBF)	132.72 (9.28) ^BC^	6.20 (0.88) ^BC^	50.00 (3.67) ^D^
	Tetric EvoCeram Bulk-Fill (TBF)	118.63 (7.76) ^BC^	5.44 (0.95) ^C^	43.80 (2.77) ^D^
	X-tra fil (XBF)	136.10 (12.46) ^B^	6.86 (0.36) ^B^	77.40 (4.72) ^B^
	Quix Fill (QBF)	115.44 (9.58) ^C^	5.48 (0.59) ^C^	69.60 (2.88) ^C^
	Grandio (GR)	215.13 (8.13) ^A^	8.50 (0.44) ^A^	90.80 (2.16) ^A^
Distilled Water Control	Filtek Bulk-Fill (FBF)	126.11 (8.91) ^BC^	5.73 (0.38) ^BC^	62.00 (6.16) ^C^
	Tetric EvoCeram Bulk-Fill (TBF)	109.99 (8.80) ^C^	4.96 (0.53) ^C^	48.40 (2.88) ^D^
	X-tra fil (XBF)	128.77 (6.86) ^B^	6.32 (0.76) ^B^	75.80 (4.49) ^B^
	Quix Fill (QBF)	109.00 (11.81) ^C^	5.27 (0.70) ^BC^	68.20 (2.59) ^BC^
	Grandio (GR)	205.83 (10.05) ^A^	8.33 (0.58) ^A^	85.00 (4.42) ^A^
Orange Juice	Filtek Bulk-Fill (FBF)	103.40 (8.36) ^BC^	3.54 (0.47) ^C^	47.60 (2.07) ^C^
	Tetric EvoCeram Bulk-Fill (TBF)	89.14 (7.92) ^C^	3.97 (0.41) ^C^	52.60 (4.16) ^C^
	X-tra fil (XBF)	108.67 (8.42) ^B^	5.01 (0.52) ^B^	75.20 (3.49) ^A^
	Quix Fill (QBF)	61.22 (8.93) ^D^	3.16 (0.58) ^C^	65.60 (1.14) ^B^
	Grandio (GR)	212.42 (7.67) ^A^	7.82 (0.59) ^A^	79.80 (1.64) ^A^
Coca-Cola	Filtek Bulk-Fill (FBF)	94.32 (8.25) ^B^	2.68 (0.45) ^B^	47.20 (3.70) ^C^
	Tetric EvoCeram Bulk-Fill (TBF)	89.19 (8.93) ^B^	3.95 (0.42) ^A^	49.40 (2.41) ^C^
	X-tra fil (XBF)	95.20 (6.67) ^B^	4.64 (0.53) ^A^	73.00 (2.00) ^A^
	Quix Fill (QBF)	50.73 (4.94) ^C^	2.33 (0.34) ^B^	56.20 (3.03) ^B^
	Grandio (GR)	132.50 (7.37) ^A^	4.53 (0.48) ^A^	72.80 (1.92) ^A^

Superscripts with dissimilar letters within the column for each separate storage condition show statistically significant differences (*p* < 0.05).

## Data Availability

The data presented in this study are available from the corresponding author upon request.

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
