# Peer review of "Mechanical and Biomimetic Characteristics of Bulk-Fill Resin Dental Composites Following Exposure in a Simulated Acidic Oral Environment"

_biomimetics, 2023, doi:10.3390/biomimetics8010019_

Round 1

Reviewer 1 Report

The theme of paper titled “Mechanical and biomimetic characteristics of bulk-fill resin dental composites following exposure in the simulated acidic oral environment” is interesting and the authors have conducted a well-designed study. However, I have a few suggestions for the authors which should be addressed.

1.      Please add the keyword ‘biomimetics’ in the list of keywords.

2.      The authors may improve the introduction and the discussion part with the help of following papers:

- Khan AA, AlKhureif AA, Mohamed BA, Bautista LS. Enhanced mechanical properties are possible with urethane dimethacrylate-based experimental restorative dental composite. Materials Research Express. 2020 Oct 16;7(10):105307.

- Khan AA, Siddiqui AZ, Mohsin SF, Al-Kheraif AA. Influence of mouth rinses on the surface hardness of dental resin nano-composite. Pakistan journal of medical sciences. 2015 Nov;31(6):1485.

3.      The references no. 27 and 28 are relatively old. Please replace these with new references.

4.      The authors have mentioned in lines 157 and 158 that “In order to prevent the accumulation of leached ingredients from the RBCs, the storage medium was repeated after every 24 hours”.  This statement should be supported with a suitable reference.

5.      How many indentations were made on each specimen for the evaluation of surface hardness? Please provide this information in section 2.4. 

Author Response

Reviewer’s Comment: The theme of paper titled “Mechanical and biomimetic characteristics of bulk-fill resin dental composites following exposure in the simulated acidic oral environment” is interesting and the authors have conducted a well-designed study. However, I have a few suggestions for the authors which should be addressed.

 Authors’ Response: The authors would like to thank the respected reviewer for reviewing and providing an insightful and constructive feedback on the manuscript that certainly helped us to improve the content. The authors worked thoroughly and revised the manuscript based on the provided suggestions. Please find a point-by-point response to the comments below: 

Reviewer’s Comment: 1. Please add the keyword ‘biomimetics’ in the list of keywords.

Authors’ Response: Thank you for your valuable suggestion, we have added the keyword ‘biomimetics’ in the list of keywords (Line 37).

Reviewer’s Comment: 2. The authors may improve the introduction and the discussion part with the help of following papers:

Khan AA, AlKhureif AA, Mohamed BA, Bautista LS. Enhanced mechanical properties are possible with urethane dimethacrylate-based experimental restorative dental composite. Materials Research Express. 2020 Oct 16;7(10):105307.

Khan AA, Siddiqui AZ, Mohsin SF, Al-Kheraif AA. Influence of mouth rinses on the surface hardness of dental resin nano-composite. Pakistan journal of medical sciences. 2015 Nov;31(6):1485.

Authors’ Response: We have incorporated both references as suggested by the reviewer (references no. 26 and 42).

 Reviewer’s Comment: 3. The references no. 27 and 28 are relatively old. Please replace these with new references.

Authors’ Response: Thank you for the valuable suggestion; accordingly, we have now replaced both references with recent references (Reference 27 and 28).

Reviewer’s Comment: 4. The authors have mentioned in lines 157 and 158 that “In order to prevent the accumulation of leached ingredients from the RBCs, the storage medium was repeated after every 24 hours”.  This statement should be supported with a suitable reference.

Authors’ Response: Based on the suggestion, a suitable reference (Ref no. 37) has been cited (Line 164).

Reviewer’s Comment: 5. How many indentations were made on each specimen for the evaluation of surface hardness? Please provide this information in section 2.4. 

Authors’ Response: A total of 3 indentations were performed for each specimen. This information has been added to the section 2.4 (Line 224).

Finally, the authors would like to thank the reviewer again for the constructive feedback and for helping the authors to improve the contents of this manuscript. We hope the quality of the manuscript has been improved and will be acceptable for publication.

Reviewer 2 Report

Dear Authors.

Congratulations on your work which, I found interesting. Manuscript: Mechanical and biomimetic characteristics of bulk-fill resin dental composites following exposure in the simulated acidic oral environment, it is well written with an adequate structure as a scientific paper demand.

I have some minor revisions to propose to you to improve your work. Please refer to the following comments:

-        Line 102:  It would be worth specifying the pH of the fluids used in the study - orange juice may have different composition, which affects the result of the study.

-        Line 151-152: Was each sample soaked in a separate container or together. Could the samples have
touched each other reducing contact with the liquid?

-        Line 165-167:  How many seconds did the polymerization last? Was the polyemrization in the middle? what about the sides of the sample, were they additionally polymerized?

-        Line 188: Were the samples measured prior to the test or were each sample assumed to be 2x2x25 mm
without measuring each sample?
-        Line 215: One measurement was made for each sample? Composite materials are inhomogeneous,
the measurement results can vary to a large extent depending on what the indenter hits.
-        Table 3 - These are repeated results from the previous table. Please consider deleting this table. -        Discussion: It is worth discussing the obtained results also with the results of standard composites
tested with similar tests (eg
Szalewski, L.; Wójcik, D.; Bogucki, M.; Szkutnik, J.;
Różyło-Kalinowska, I. The Influence of Popular Beverages on Mechanical Properties of
Composite Resins. 
Materials 2021, 14, 3097. https://doi.org/10.3390/ma14113097).
It is also worth mentioning the influence of the type of polymerization on the mechanical
properties of composite materials (e.g.
Szalewski, L.; Wójcik, D.; Sofińska-Chmiel, W.;
Kuśmierz, M.; Różyło-Kalinowska, I. How the Duration and Mode of Photopolymerization
Affect the Mechanical Properties of a Dental Composite Resin. 
Materials2023, 16, 113.
https://doi.org/10.3390/ma16010113
  ).

-        The literature is old, with many citations of works from before 2012 - please consider updating it. 

Author Response

Reviewer’s Comment: Congratulations on your work which, I found interesting. Manuscript: Mechanical and biomimetic characteristics of bulk-fill resin dental composites following exposure in the simulated acidic oral environment, it is well written with an adequate structure as a scientific paper demand. I have some minor revisions to propose to you to improve your work. Please refer to the following comments:

 Authors’ Response: The authors would like to thank the respected reviewer for reviewing and providing an insightful and constructive feedback on the manuscript that certainly helped us to improve the content. The authors worked thoroughly and revised the manuscript based on the provided suggestions. Please find a point-by-point response to the comments below: 

Reviewer’s Comment: Line 102:  It would be worth specifying the pH of the fluids used in the study - orange juice may have different composition, which affects the result of the study.

Authors’ Response: Thank you for the important suggestion. We have now mentioned the pH of Coca-Cola and Orange Juice in section 2 (Lines 104-105).

Reviewer’s Comment: Line 151-152: Was each sample soaked in a separate container or together. Could the samples have touched each other reducing contact with the liquid?

 Authors’ Response: The specimens were placed in a way that they did not touch each other and each specimen got similar exposure to the corresponding medium. This information has been added to the manuscript for further clarity (Lines 159-161).

Reviewer’s Comment: Line 165-167:  How many seconds did the polymerization last? Was the polyemrization in the middle? What about the sides of the sample, were they additionally polymerized?

Authors’ Response: The specimens were polymerized in an orbital sequence four times for 20 s each in overlapping shots. This information and related citation has been added (Lines 173-175).

Reviewer’s Comment: Line 188: Were the samples measured prior to the test or were each sample assumed to be 2x2x25 mm without measuring each sample? -      

Authors’ Response: The dimensions of samples were measured prior to storage in immersion media (Lines 152-54).

Reviewer’s Comment:  Line 215: One measurement was made for each sample? Composite materials are inhomogeneous, the measurement results can vary to a large extent depending on what the indenter hits. -        

Authors’ Response: A total of three indentations were performed for each specimen and this information has been added in section 2.4 (Line 224).

Reviewer’s Comment: Table 3 - These are repeated results from the previous table. Please consider deleting this table.

Authors’ Response: The authors agree with the reviewer’s point that there is overlap in the data presented in table 2 and 3. As the purpose is to highlight the significance of material type and storage conditions separately, therefore we believe the presentation of results in this way will enhance the understanding of readers.

Reviewer’s Comment: It is worth discussing the obtained results also with the results of standard composites tested with similar tests (eg Szalewski, L.; Wójcik, D.; Bogucki, M.; Szkutnik, J.;
Różyło-Kalinowska, I. The Influence of Popular Beverages on Mechanical Properties of
Composite Resins. Materials 202114, 3097. https://doi.org/10.3390/ma14113097).

Authors’ Response: Based on the suggestion, we have now added a comparison of our results with the suggested study (Ref no: 49) in the discussion section (lines 381-387).

Reviewer’s Comment: It is also worth mentioning the influence of the type of polymerization on the mechanical
properties of composite materials (e.g. Szalewski, L.; Wójcik, D.; Sofińska-Chmiel, W.;
Kuśmierz, M.; Różyło-Kalinowska, I. How the Duration and Mode of Photopolymerization
Affect the Mechanical Properties of a Dental Composite Resin. Materials202316, 113.
https://doi.org/10.3390/ma16010113  ).

Authors’ Response: Based on the suggestion, we have now added brief description of the suggested study (Ref no: 50) in the discussion section (lines 388-392).

Reviewer’s Comment: The literature is old, with many citations of works from before 2012 - please consider updating it. 

Authors’ Response: As suggested, the authors have included a number of recent references (last 3 years) as highlighted in the references list (References 1-3, 8, 12, 24-32, 34, 38, 43, and 49-51).

Finally, the authors would like to thank the reviewer again for the constructive feedback and for helping the authors to improve the contents of this manuscript. We hope the quality of the manuscript has been improved and will be acceptable for publication.